# Characterization of the Time–Space Evolution of Acoustic Emissions from a Coal-like Material Composite Model and an Analysis of the Effect of the Dip Angle on the Bursting Tendency

Pengxiang Zhao [1,2,3,4,5], Jian Wen [1,*], Shugang Li [1], Weidong Lu [4,5], Yongchen He [6], Fang Lou [3] and Laolao Wang [2]

1   School of Safety Science and Engineering, Xi'an University of Science and Technology, Xi'an 710054, China; zhpxhs@sina.com (P.Z.); lisg@xust.edu.cn (S.L.)
2   Key Laboratory of Xinjiang Coal Resources Green Mining, Ministry of Education, Xinjiang Institute of Engineering, Urumqi 830023, China; wll196295@163.com
3   Xinjiang Coal Science Research Institute, Urumqi 830091, China; 15593529614@163.com
4   School of Safety Science and Engineering, Xinjiang Institute of Engineering, Urumqi 830023, China; g15829465683@163.com
5   Xinjiang Key Laboratory of Coal Mine Disaster Intelligent Prevention and Emergency Response, Xinjiang Institute of Engineering, Urumqi 830023, China
6   School of Energy and Power Engineering, Xi'an Jiaotong University, Xi'an 710049, China; 17358163173@163.com
*   Correspondence: 17403070431@stu.xust.edu.cn

**Abstract:** Rock bursts pose a grievous risk to the health and lives of miners and to the industry. One factor that affects rock bursts is the dip angle of the coal seam. Because of the uniquely high gas content of the coal in a mine in Shanxi Province, China, coal specimens were obtained from this mine to produce coal–rock combination specimens and test the effects of various seam inclinations. Using a DYD-10 uniaxial compression system and a PCI-8 acoustic emission (AE) signal acquisition system, we investigated the spatial and temporal evolution characteristics of the burst tendency of specimens with different coal seam inclination angles (0°, 10°, 20°, 30°, 35°, 40°, and 45°). Uniaxial pressure was applied to the specimens, and we found that, as the inclination angle increased, the coal–rock combination specimens exhibited structural damage and destabilization, which was attributed to the generation of an interface slip phenomenon. In all tests, the coal exhibited greater damage than the rock. There was an energy convergence at the coal–rock interlayer interface, which was the main carrier for the accumulated energy. The impact energy dissipation index is defined according to the energy dissipation properties of the loading process of coal–rock composites. As the inclination angle increased, the impact energy dissipation index, energy storage limit, compressive strength, elastic modulus, and other indexes gradually decreased. This effect was strongest where the angles were 40° and 45°. The indexes used to assess the impact propensity decreased to a notable degree at these angles, revealing that the burst tendency of coal–rock is curtailed as the inclination angle increases. The results of this research are of great importance to the early evaluation of mine burst risks and the sustainable development of coal utilization.

**Keywords:** spatial and temporal evolution characteristics; coal seam inclination angles; coal–rock combination specimens; impact energy dissipation index; energy storage limit

## 1. Introduction

China is undertaking the intensive mining of deep coal seams, and as the mines become deeper, increasingly high-strength approaches are required to mine the coal. This deep, high-strength approach increases the hazards associated with mining, leaving miners subject to rock bursts and other potential disasters [1–4]. Under the influence of coal mining, the main reason for the destabilization and fracture of coal seams and rock seams is the sudden release of their own elastic energy, which leads to the deformation and slippage of

the combined coal–rock structure [5–7]. Thus, it is vital to study the fracture mechanisms of coal–rock combinations under static loading conditions to ensure the sustainability of coal mining and to decrease the number of rock burst accidents.

Coal seams are widely distributed around the world, and the conditions of their occurrence are complex. In order to research the mechanical characteristics of coal–rock combinations during the failure process, many scholars have conducted extensive laboratory experiments to analyze the fracture evolution laws of the specimens [8,9]. Li et al. [10] researched the mechanical characteristics of coal–rock combinations during uniaxial compression, and the results showed that the combined strength of the coal–rock combinations was higher than that of a single coal body. Wang et al. [11,12] researched the failure mechanism of coal–rock composites under various confining pressure conditions and compared it with that of a single coal body; they found that the confining pressure effect had an important influence on the formation of the final failure mode of the combinations. As the pressure increased, the number of fissures increased significantly, and the fracture effect intensified, exhibiting a mixed tensile and shear failure mode. Jin et al. [13] carried out grouting tests on crushed coal–rock and examined the failure characteristics of grouting specimens under uniaxial compression conditions. They found that the modulus of the elastic and peak axial strain of the grouting specimens was altered by changes in the solidified grout strength (SGS). Yin et al. [14] analyzed the uniaxial and triaxial loading processes of coal–rock combinations and a single prefabricated crushed coal specimen under different inclination angles; the results demonstrated that lithology fundamentally affects the strength, macro failure initiation (MFI), and fracture characteristics of the material. Li et al. [15] carried out permeability stress tests on coal–rock composites and analyzed the influence of different factors on the permeability of the combinations. It is of great significance to study the permeability variation characteristics of coal–rock composites.

During the process whereby coal–rock combinations fracture under uniaxial compression, the internal energy is released outward in the form of elastic energy, which generates AE signals. The changes in the parameters of the acoustic emission characteristics can reflect the evolution and fracture characteristics of internal deformation in coal–rock combinations. Therefore, AE technology is widely used to study the fracture characteristics of coal–rock combination specimens [16–18]. Song et al. [19] conducted uniaxial compression experiments to analyze the failure characteristics of coal–rock combination specimens using the variation rules of the AE count rate and peak frequency; their results showed that the acoustic emission technology has a superior ability to analyze the fracture characteristics of coal–rock combinations. Liang et al. [20] researched the fracture characteristics of soft rock-coal combinations under different loading conditions and found that the AE energy and AE count decreased with an increase in the proportion of soft rock. Zhang et al. [21] analyzed the fracture behavior of coal–rock combinations with different water contents under uniaxial compression using AE technology; they found that the AE counts showed different forms of decreasing with the expansion of cracks in the specimens.

In recent years, many scholars have conducted extensive research on the impact tendencies of coal and rock, and many indices have been proposed to judge them. These include the elastic energy index, the elastic modulus, the uniaxial compressive strength, and the remainder energy index [22–24]. Zhang et al. [25] researched the effect of different retreat speeds on the impact tendency of coal, and the results showed that high retreat speeds have a driving effect on the impact tendency.

In summary, the existing research mainly focuses on the fracture characteristics of coal–rock combinations under different coal thickness ratios. However, limited research has considered different coal seam dip angles. This article takes the coal–rock combination as the research model, prepares coal–rock combination specimens with different coal–rock interface inclination angles, and carries out research on the AE and impact tendency characteristics of coal–rock combinations under different coal–rock interface inclination angles. The research provides a basis for the comprehensive prevention and technological

control of rock bursts; it is of great importance to realize safe coal mine production and the sustainable development of coal resource exploitation.

## 2. Experiment

### 2.1. Specimen Preparation

This experiment took place in the geological conditions of a high-gas mine in Heshun, Shanxi, PR China. The mine coal seam in question is gently inclined (0°–25°). The geographical location of the mine, as well as the thickness, strength, and related mechanical parameters of the coal seam (0°–25°), are shown in Figure 1 and Table 1. The complexity of the geological conditions made it necessary to assess which factors to include. This experiment included seven coal–rock interface inclinations (0°, 10°, 20°, 30°, 35°, 40°, and 45°). The rock–coal–rock strength ratio was determined to be 3:1:3 (as shown in Figure 2 and Table 2). This meant testing seven sets of symmetrical composite standard cylindrical specimens (specimen size: 50 mm × 100 mm), with 35 specimens in total. To reduce errors between the model and the actual physical engineering parameters, this article chose coal-like materials to create the coal–rock combination samples. For the rock part of the coal-rock combination samples, we used sand, cement, and plaster for production. Among them, the mass of sand was 1200 g, the mass of cement was 180 g, and the mass of plaster was 216 g. This standardized the analysis of the AE characteristics and impact tendency.

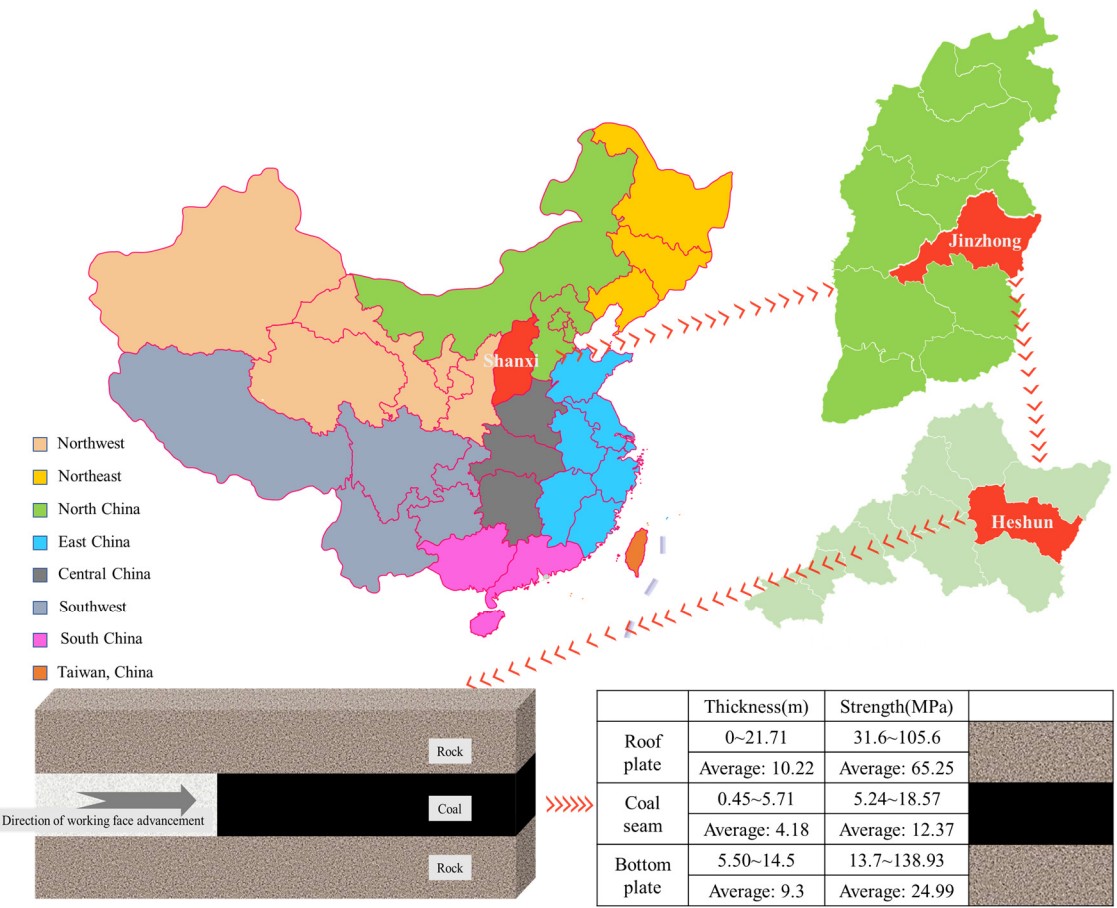

|  | | Thickness(m) | Strength(MPa) |  |
|---|---|---|---|---|
| Roof plate | | 0~21.71 | 31.6~105.6 | |
| | | Average: 10.22 | Average: 65.25 | |
| Coal seam | | 0.45~5.71 | 5.24~18.57 | |
| | | Average: 4.18 | Average: 12.37 | |
| Bottom plate | | 5.50~14.5 | 13.7~138.93 | |
| | | Average: 9.3 | Average: 24.99 | |

**Figure 1.** Geographical location and geological characteristics of the mine.

**Table 1.** Mechanical parameter characteristics of coal seams.

| Mechanical Parameters | Bulk Density (kN·m$^{-3}$) | Poisson's Ratio | Cohesive Force (MPa) | Dilatancy Angle (°) | Internal Friction Angle (°) |
|---|---|---|---|---|---|
| Coal seam | 14.6 | 0.275 | 2.9 | 8 | 20 |

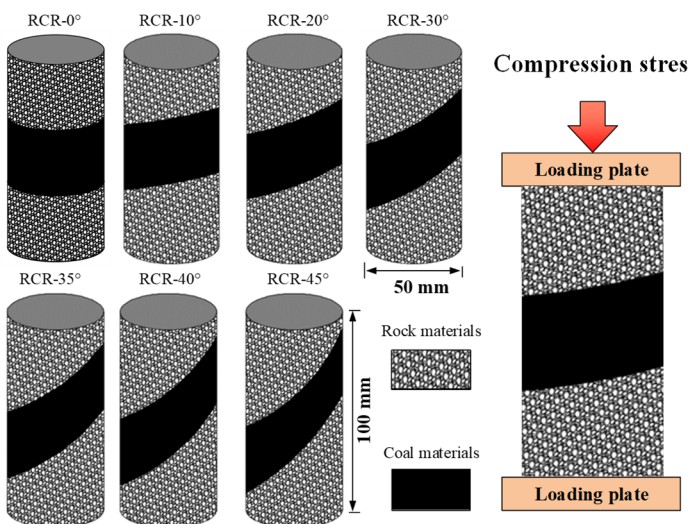

**Figure 2.** Diagram of the coal–rock combination specimens (0°–45°) under uniaxial compression.

**Table 2.** Various characteristics of the coal–rock material ratio, tested using a DYD-10 uniaxial compression system.

| Proportions | Coal Powder (%) | Cement (%) | Plaster (%) | Compressive Strength (MPa) |
|---|---|---|---|---|
| Rock materials | 0 | 15 | 18 | 1.33 |
| Coal materials | 35 | 10 | 6 | 0.43 |

Note: the Weihe sand was aggregated, the mass was fixed at 1200 g, and the material percentage (%) was calculated as the fixed Weihe sand mass.

### 2.2. Experimental System

The experimental system was composed of a DYD-10 uniaxial compression system and a PCI-8 AE signal acquisition system (Figure 3). The DYD-10 uniaxial compression system was composed of a load frame, a servo motor, a transmission system, a data acquisition system, and a displacement protection device; the maximum load capacity was 10 KN. The AE signal acquisition system was composed of sensors, a preamplifier, an acquisition box, and a computer host. The AE sensor model was NANO-30 (the sensor were manufactured by Hunan Enditi Technology Co., Ltd. in Changsha, China), the operating frequency range was 125~750 kHz, the resonant frequency was 140 kHz, and the peak frequency was 300 kHz. Each group of specimens included three standard specimens for the AE characteristics experiment so that the holistic deformation and failure processes could be examined under repeated uniaxial static loads.

### 2.3. Test Procedure

Step 1: The surfaces of the prepared specimens were smoothed so that the roughness did not exceed 0.02 mm, so as to enhance the accuracy of the experimental results.

Step 2: The loading mode of the DYD-10 uniaxial compression system was set to displacement-controlled loading, and the loading rate was 0.005 mm/s. In addition, it was necessary to debug the AE signal acquisition system multiple times; the AE system threshold was set to 40 dB and the sampling frequency was 1 MHz. Each channel corresponded to an independent AE sensor and preamplifier, and the signal magnification was 40 times.

Step 3: The acoustic emission sensors were arranged in a ring on the surface of the specimen (Figure 3). Meanwhile, in order to prevent the sensors from slipping, coupling agents were added at the contact point between the sensors and the specimens.

Step 4: The specimen was placed on the central axis of the lower pressure plate of the DYD-10 uniaxial compression system, and the handle was used to adjust the upper pressure plate of the system to tightly fit the specimen while ensuring the provision of sufficient light in the experimental environment.

Step 5: Before the experiment began, it was necessary to conduct a lead-breaking test to ensure that the AE sensors were well connected. During the lead break test, the surrounding environment was kept quiet to prevent noise from affecting the capture of AE signals.

Step 6: The specimen was loaded using the uniaxial compression system; when the loading pressure appeared on the monitor, the AE signals began to be collected and recorded by the high-definition camera. Then, when the compression stress was reduced to sixty percent of the compressive strength, the loading was stopped. The AE data were then collected and analyzed.

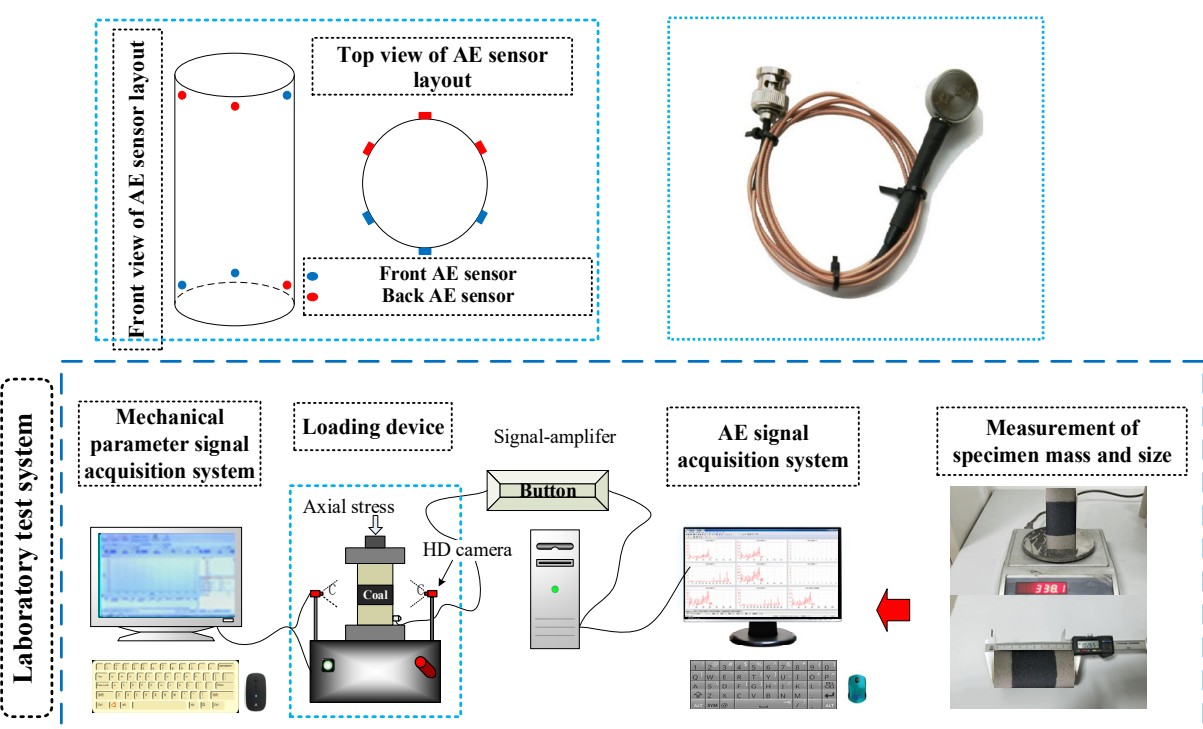

**Figure 3.** Schematic diagram of the experimental system.

## 3. Experimental Results

### 3.1. Stress–Strain Curve and Fracture Form of Specimens under Uniaxial Compression

Figure 4 shows the stress–strain curves of the specimen under compression conditions at different angles of inclination. Furthermore, some composite specimens and final failure characteristics are shown in Figure 5. Judging from the experimental data for the angles 0° and 10°, both specimens had relatively high levels of compressive strength. However, because the strength of the coal body was less than that of the surrounding rock, as the compression stress increased, the failures first seen in the coal led to subsequent failures up and down the specimen as a whole. The sustainable axial stress eventually reached a peak; then, the specimen exhibited continuous and complete penetration.

When the coal seam inclination angle was 20°, a clamping effect between the upper and lower rock materials was observed. The cracking then spread to the lower parts of the specimen. When the coal–rock interface inclination angles were 30° and 35°, the effect of

inclination grew larger. Under the influence of coal seam failure, the coal–rock interface became unstable, leading to a rise in the number of rock cracks. When the inclination angle was 40°, emergent cracks appeared to be offset in the direction of the inclination angle. When the inclination angle was 45°, the combination specimens illustrated obvious slippage and coal body spalling. The breakdown in the coal–rock composite was not due to the destruction of the coal specimen, as it attained its strength limit instead of the coal–rock interface. The structural slippage created its own instability.

In order to further analyze the characteristics of the fracture modes of the coal quantitatively in rock combination samples under different inclination angles, the peak strain of the samples with different inclination angles was obtained, as shown in Figure 6. As the inclination angle rose, the peak strain raised first (0~40°) and then decreased (45°); this was because, during the process of specimen failure, the fracture mode was gradually changed from splitting fracture to slip fracture. When the inclination angle ranged from 0° to 40°, the fracture mode of the specimen was that of a splitting fracture. When the inclination angle was 40°, the peak strain was the largest; this was the turning point from the splitting fracture mode to the slip fracture mode. When the inclination angle was 45°, the specimen did not yield during the uniaxial compression process, and the interface of the coal–rock composite specimen slipped under the external force, resulting in the structural slip fracture of the specimen as a whole; the peak strain at this time was $16.75 \times 10^{-3}$.

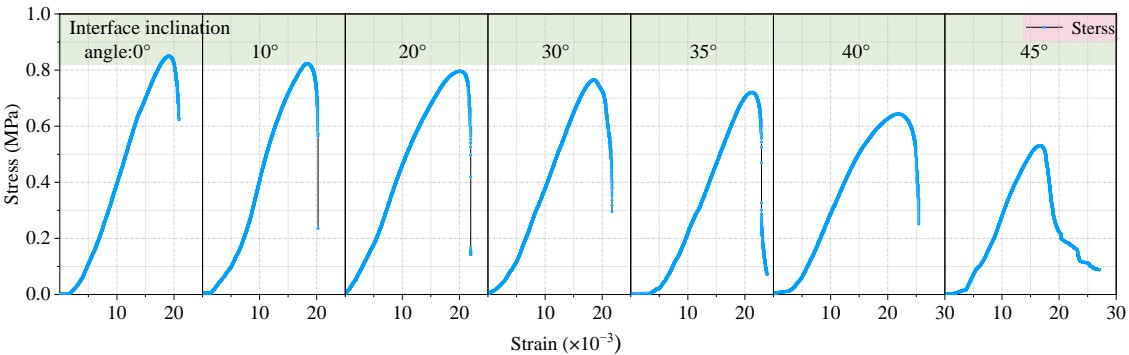

**Figure 4.** Stress–strain curves for specimens under various coal seam inclination angles.

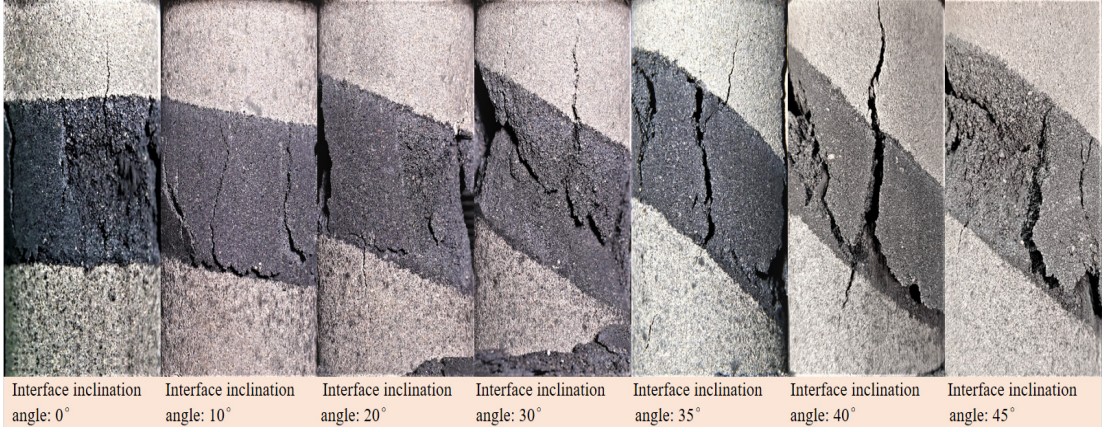

**Figure 5.** Fracture modes of the tested samples for different rock–coal inclinations.

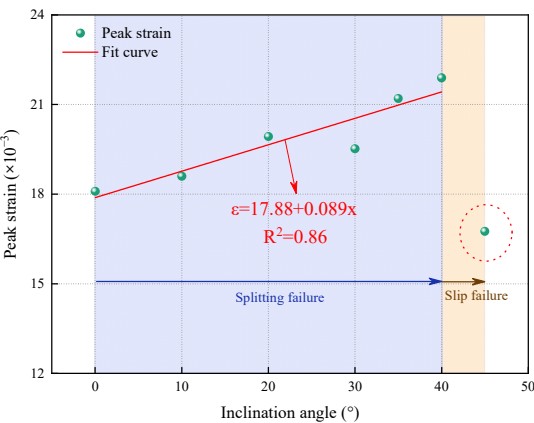

**Figure 6.** Variation in the peak strain at different inclination angles.

### 3.2. Energy Evolution Characteristics of Specimens under Uniaxial Compression

In their natural state, coal–rock bodies are affected by nearby mining disturbances, which cause them to exhibit characteristics such as small cracks and fissures. These can lead to expansion, perforation, or macroscopic deformation, and then slippage and collapse. There are numerous factors influencing this phenomenon, such as energy accumulation, restoration, stability, and violent release. The two main styles of energy transfer that occur under the influence of an external force are unidirectional irreversible energy and the elastic energy that exists in specimens. Assuming that the experimental environment is a closed system, according to the first law of thermodynamics, it can be deduced that the total energy input from outside must satisfy the following relationship.

The total input energy from outside during the uniaxial compression process $W$ can be obtained using Equation (1), and the reversible elastic energy $W_e$ can be obtained from Equation (2). According to the conservation of energy, the dissipated energy $W_d$ can then be calculated according to Equation (3) [26]:

$$W = \int \sigma_1 d\varepsilon_1 = \sum_{i=1}^{n} \frac{1}{2}(\sigma_{i+1} + \sigma_i)(\varepsilon_{i+1} - \varepsilon_i) \tag{1}$$

$$W_e = \frac{1}{2}\sigma_1\varepsilon_1 = \frac{\sigma_1^2}{2E_i} \tag{2}$$

$$W_d = W - W_e \tag{3}$$

where $W$ is the total energy generated by the axial load on the composite specimens, $kJ/m^3$; $W_e$ is the reversible elastic energy that exists in the specimens, $kJ/m^3$; $W_d$ is the energy irreversibly dissipated during specimen failure, $kJ/m^3$; $\sigma_i$ is a random point on the stress–strain curve, MPa; $\varepsilon_i$ is the strain in response to the point, and the original value is 0; and $E_i$ is the unloading modulus of elasticity. For the convenience of calculation, generally, the initial modulus of elasticity $E_t$ is equated to $E_i$.

The relevant data from the tests were run through Equations (1)–(3) to obtain each energy value during the compression tests. These equations were then used to compute the failure point and combined stress–strain curves that showed the energy dissipation characteristics of the materials. Because of the limited space available, only the coal–rock interface data for a 20° inclination is presented here.

Figure 7 shows that, as the strain increased, the total energy input from outside and the dissipated energy were nonlinear, with the elastic energy first increasing and then decreasing. Before the specimen was unstably broken, the elastic strain was the dominant factor. The external input energy mainly transferred the elastic strain energy to the interior of the specimen. Under these conditions, the specimen was unable to dissipate that energy. This means that, when approaching failure, the AE signal received was low and unsteady. Meanwhile, the rate of increase in the energy that the specimen could accumulate was

low. The press successfully produced the pressure necessary to develop and propagate internal cracks, showing the subsequent fracture process. When cracking occurred, this led to prompt energy dissipation, whereby the energy growth rate increased sharply and cracks expanded quickly. Numerous AE signals were received during these events, and the AE energy increased sharply. The cracking events formed peaks; thereafter, the dissipation energy continued to increase, but more gradually, until the remaining energy was released. Finally, the specimen reached a critical failure point and became unstable. Therefore, in Figure 7, OA represented the stage of slow energy storage, AB represented the stage of elastic energy growth, BC represented the stage of energy dissipation, and CD represented the stage of residual energy release.

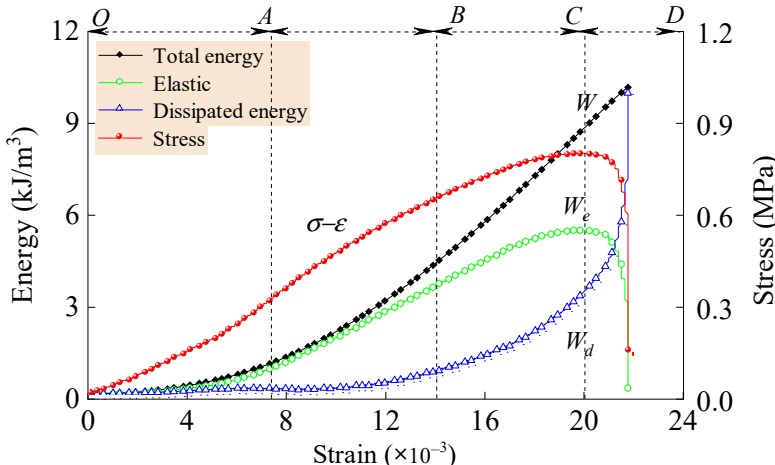

**Figure 7.** Energy dissipation characteristics during the loading process for an angle of 20° at the coal–rock interface.

## 4. Discussion

### 4.1. Crack Propagation and Characteristics of the Temporal and Spatial Evolution of AE

Figures 8 and 9 show the coal–rock assemblage specimens and their interface inclination angles, which, in this case, are 20° and 45°, respectively. Demonstrating the different failure characteristics involved testing the uniaxial compression and recording the process using a camera; furthermore, the specimens were also tested in an inverted state.

When the rock–coal interface inclination angle was 20° and the cracks expanded, the evolution of the cracks followed the three-dimensional AE event figures, as shown in Figure 8. This was the same as for the other coal–rock composite specimens. There were few acoustic emission positioning events during the early loading phase ($\sigma = 0.213$ MPa, $\varepsilon = 0.0057$); however, as the initial cracks occurred ($\sigma = 0.616$ MPa, $\varepsilon = 0.0134$), they appeared slightly offset and then expanded along the direction of the seam. Meanwhile, the number of AE event locations remained concentrated at the rock–coal interface, and the cracks expanded at that interface. As the axial stress increased gradually ($\sigma = 0.727$ MPa, $\varepsilon = 0.0148$), crack expansion accelerated, which led to the extension of a fissure in the lower section of the coal–rock composite; this was accompanied by AE events. The stress peak was reached at ($\sigma = 0.79$ MPa, $\varepsilon = 0.0199$), and this event can be seen in the AE location figure. Thus, there were dense AE location events in the right part of the coal seam. Cracking occurred in this interior section of the specimen, and the width of the crack increased, generating notable AE location events. After the peak ($\sigma = 0.506$ MPa, $\varepsilon = 0.0223$), cracks appeared throughout the entire coal body, and coal lumps broke off. The interior crack had extended down the middle of the rock, along with the rock–coal interface at the lower end. At this point, the AE events reached their maximum.

For the 45° specimen, the cracks appeared as shown in Figure 9. In this specimen, the compressive strength was low compared with that of the other specimens included in this experimental group. Initially, the specimens did not break, and there were few AE signals

and positioning events. When the axial stress increased to $\sigma = 0.329$ MPa ($\varepsilon = 0.01091$), the first accumulative stress events were detectable as an AE event, as were minor exterior cracks. The initial cracks appeared along the inclination path ($\sigma = 0.4558$ MPa, $\varepsilon = 0.0148$), being caused by the axial load. The compression friction of the rock–coal interface then led to numerous AE events. When the axial stress reached a peak ($\sigma = 0.53$ MPa, $\varepsilon = 0.01604$), myriad cracks appeared and expanded across the seams, leading to a large surface crack. AE location events increased sharply, but only a small effect was externally visible. The 45° specimen set was nearly rendered invalid due to the high levels of slippage; the sustainable peak strain of this specimen was exceptionally low. After the peak ($\sigma = 0.1002$ MPa, $\varepsilon = 0.02558$), the axial stress increased suddenly, leading to a reduction in stress and aggravating the failure level of the coal seams; and the specimen proceeded to break apart. In comparison with the lower-inclination specimens (i.e., 20°), the relative instability of the 45° specimen was clear.

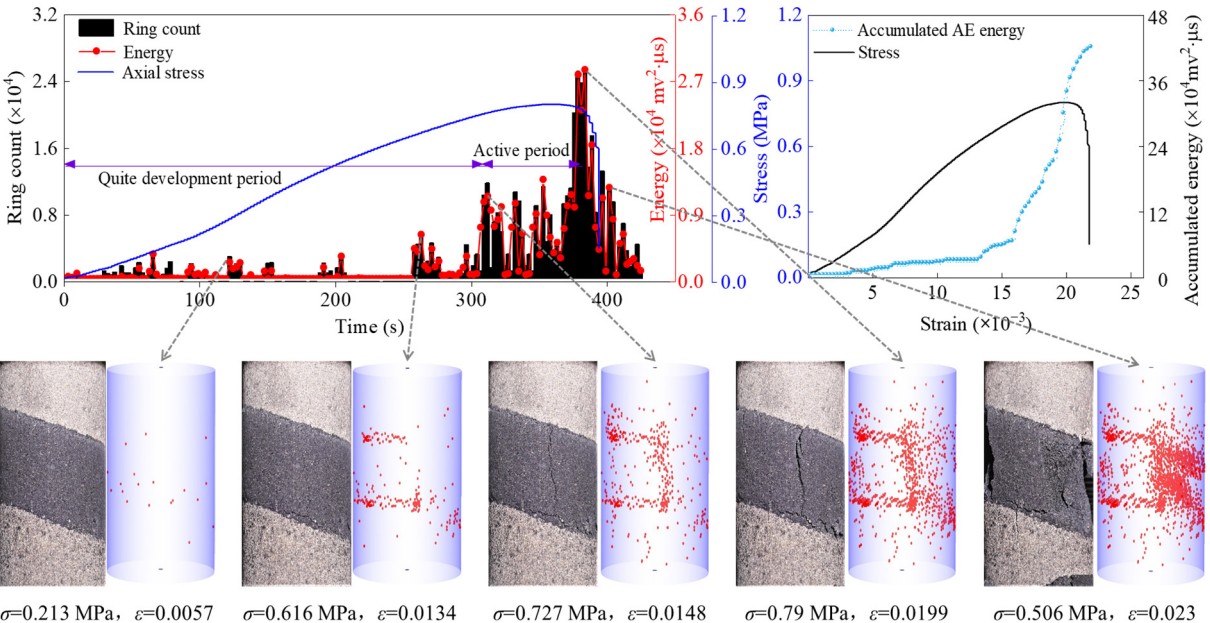

**Figure 8.** Crack propagation and AE characteristics of the specimens with an incidence of 20°.

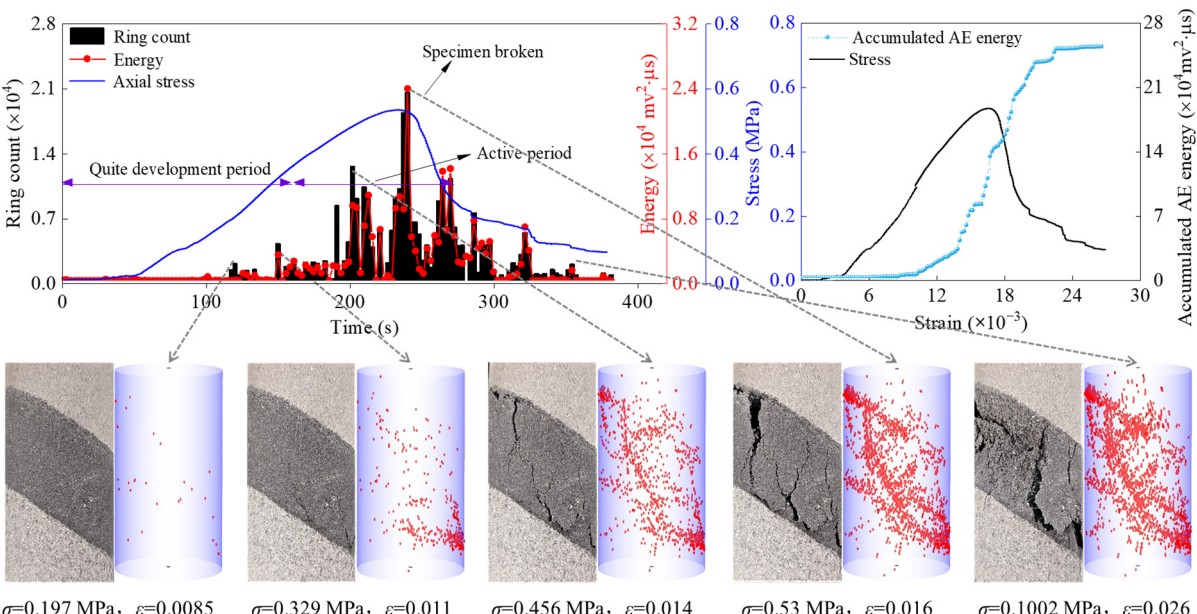

**Figure 9.** Crack propagation and AE characteristics of the specimens with an incidence of 45°.

When analyzing the AE energy characteristics of the coal–rock combinations during the fracture process, the frequency characteristics were also an important parameter used to assess the mechanical behavior of the coal–rock composite specimens during the fracture process, and the mechanical states of the specimens at different time periods can be further understood via an analysis of frequency characteristics [27,28]. Because the simple harmonic vibration corresponding to the peak frequency can restore the original acoustic signal of the specimens to the greatest extent, the peak frequency was selected as the characteristic parameter of the AE frequency. Figure 10 shows the change rule of the peak frequency for the interface inclination angles of 20° and 45°. Moreover, according to [29], the peak frequency range of 10~80 kHz was defined as the low-frequency band, 80~150 kHz was defined as the middle-frequency band, and values above 150 kHz were defined as the high-frequency band.

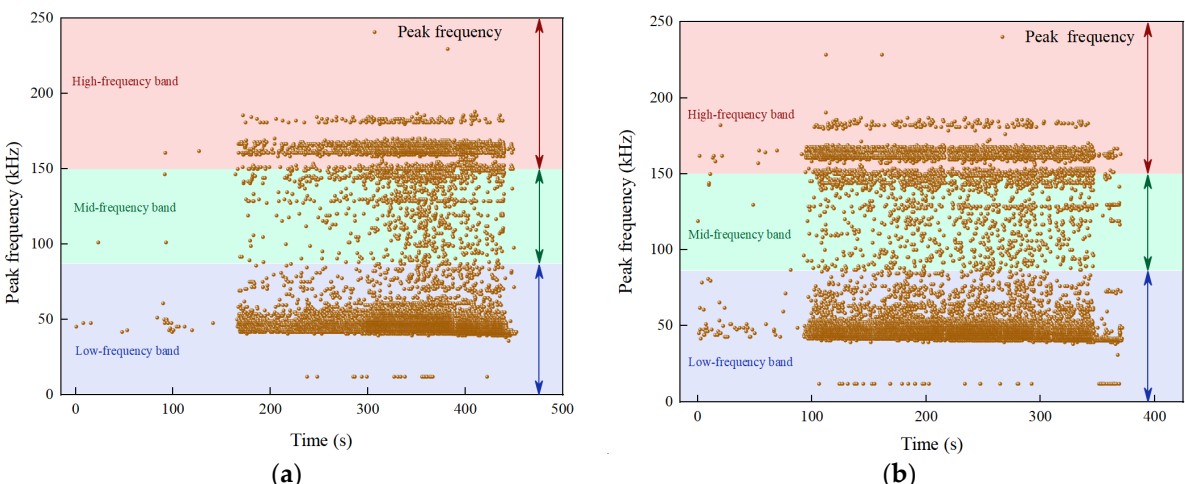

**Figure 10.** Variation characteristics of the peak frequency under different inclination angles. (**a**) 20°; (**b**) 45°.

Figure 10 shows that, for the two inclination angles, low-frequency signals were mainly centralized in the range of 40–70 kHz, mid-frequency signals were mainly centralized in the range of 140–150 kHz, and high-frequency signals were mainly centralized in the range

of 160–170 kHz. During the early stage of loading, the peak frequencies were mainly centralized in the low-frequency band; as the loading time increased, cracks gradually occurred in the specimens, and the mid-frequency and high-frequency signals appeared successively. Meanwhile, as the degree of sample fracture gradually increased, energy was gradually released, leading to a corresponding increase in high-frequency signals. When the inclination angle was 45°, the energy release rate accelerated significantly, and, compared to the inclination angle of 20°, the degree of damage was more obvious, meaning that the peak frequency performance was more active.

*4.2. Influence of the Interface Dip Effect on the Impact Tendency and Fracture Development*

When mining coal seams, breaks in the coal–rock body are usually induced by the release of intrinsically stored elastic energy, which leads to rock bursts. Therefore, understanding the mechanism of crack propagation and its relationship with the inclination angle helps to determine the impact tendency. Scholars have incorporated various factors into their analyses of the impact tendency, such as material dynamics, failure times, impact energy indices, coal–rock body compressive strengths, and critical softening area coefficients. During the process of the uniaxial compression of coal–rock combinations, there is a pre-peak elastic energy accumulation phase, and the fracture of the coal–rock body leads to a sharp increase in the dissipating energy during the post-peak stage. Based on the energy dissipation peculiarities of the coal–rock composites' uniaxial compression process, it can be determined that the impact energy dissipation index is the ratio of the elastic energy accumulated before the peak and the dissipated energy after the peak during the uniaxial compression of the coal–rock body. The impact energy dissipation index reflects the dynamic change in energy accumulation as well as the internal energy release within the sample, and it can be expressed by Equation (4) [30]:

$$K = \frac{U_1}{U_2} \tag{4}$$

where $K$ is the impact energy dissipation index, $U_1$ is the elastic energy accumulated pre-peak in kJ/m$^3$, and $U_2$ is the dissipated energy released post-peak in kJ/m$^3$.

Figure 11 shows the correlation between the impact energy dissipation index, the energy storage limit, the compressive strength, and the elastic modulus of various specimen inclination angles. Of these, the energy storage limit was used to analyze the elastic energy storage potential under real-world conditions. Generally, $\sigma_c^2/(2E)$ is used to calculate the energy storage limit, and $\sigma$c and $E$ are the indices of the compression strength and elastic modulus, respectively. Figure 11 also shows that, as the seams' inclination angle increased, the impact energy dissipation index, energy storage limit, compression strength, and elastic modulus exhibited a decreasing tendency. This was especially true for higher inclination angles, such as 40° and 45°. The indicators used to assess impact propensity were greatly reduced.

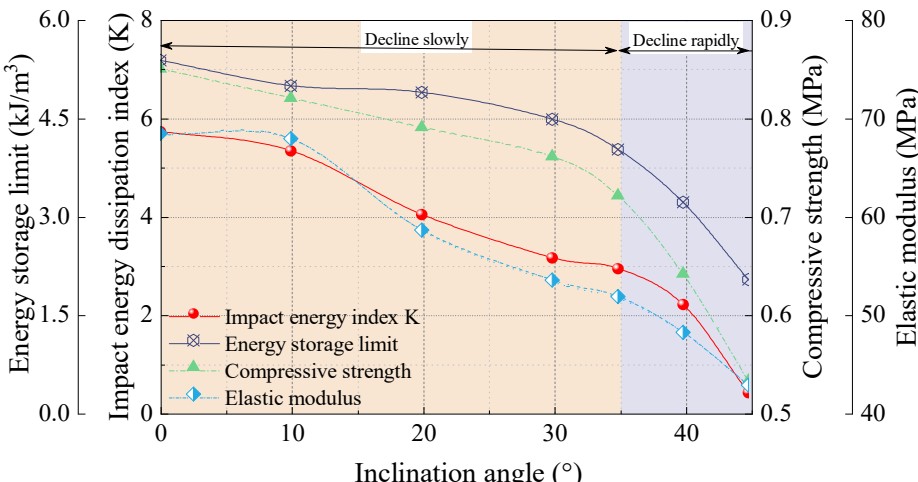

**Figure 11.** Energy storage limit, impact energy dissipation storge, compressive strength, and elastic modulus of the specimens with the change in inclination angles.

Based on an examination of the failure characteristics, the evolution of AE location events, and the energy accumulation indices, the coal seam section of the composite was clearly the main carrier for energy accumulation. It was also the point that first exhibited failure, where cracks developed and eventually undermined the stability of the composite specimen. As the inclination angle increased, the specimen's carrying ability was reduced, and the entire composite's stability was diminished, curtailing the energy storage limit. This had the effect of releasing the latent energy, and the possibility of composite impact was decreased. When the interface inclination angles were 40° and 45°, the axial stress could not achieve a strength level sufficient to determine the stress–strain process. The coal–rock interface demonstrated slippage when it came under external pressure; this led energy to be released through the friction interface, meaning that the energy was no longer stored internally. The slippage effectively alleviated the pressure, such that the impact tendency of the combination decreased as the inclination increased.

## 5. Conclusions

In this study, uniaxial compression tests and AE experiments were carried out on coal–rock combinations with different coal–rock interface inclination angles, and the mechanical behaviors and AE characteristics of the combinations were investigated during the fracture process. In addition, relevant energy parameters were introduced to research the energy evolution characteristics of the specimens. The main conclusions of this paper are as follows:

(1) As the interface inclination angle increases, the coal seam in the coal–rock combination becomes unstable and fractures due to the interface slip phenomenon; this causes the rock to fracture to a certain extent. Moreover, the compressive strength and elastic modulus of the coal–rock combination show a declining trend as the inclination angle increases.

(2) When the combined body fractures, the degree of destruction of the coal is higher than that of the rock, and the number of AE location events is also larger. Therefore, it can be determined that coal is the main source of energy accumulation in the combination and also the main carrier of energy accumulation. The overall properties of the combination are mainly determined by the coal seam.

(3) As the interface inclination angle raises, the bearing capacity and deformation resistance of the coal–rock combination are reduced to a certain extent, resulting in a decrease in the energy storage limit and energy release speed. At the same time, the impact energy index also shows a downward trend. In particular, when the inclination angle is 40° or 45°, this downward trend increases significantly.

(4)  This article conducts an in-depth analysis of the failure characteristics of coal-rock combinations under uniaxial compression conditions at different inclination angles, which is important for researching the fracture behavior of coal rock combinations. However, there are still some limitations. This article only conducts research from the perspective of physical experiments, without combining numerical simulation methods to study the fracture characteristics of coal-rock combinations. Based on the deformation and slip characteristics of coal rock combinations, discrete element analysis can effectively simulate their failure modes. The results are not only more accurate, but they also provide certain guidance for physical experiments. Therefore, in future research, we plan to use discrete element analysis as an important research method to further study the fracture characteristics of coal-rock combinations under uniaxial compression conditions.

The results of this study are of great significance to the further study of the fracture mechanisms of coal–rock combinations under uniaxial compression conditions; the findings presented here will make a significant contribution to promoting sustainable coal utilization.

**Author Contributions:** Conceptualization, P.Z.; methodology, S.L.; data curation, J.W. and Y.H.; writing—original draft preparation, J.W.; investigation, W.L.; resources, F.L. and L.W.; writing— review and editing, Y.H. and F.L.; funding acquisition, W.L. and L.W. All authors have read and agreed to the published version of the manuscript.

**Funding:** This research was funded by grants from the National Natural Science Foundation of China (5217-4205, 5197-4237, and 5207-4217), the Shanxi Province Outstanding Youth Science Fund Project (2023-JC-JQ-40), the Key Research and Development Task Special Sub-Project of Xinjiang Uygur Autonomous Region (2022B01034-3), and the Project of Key Laboratory of Xinjiang Coal Resources Green Mining, Ministry of Education (KLXGY-KA2404). The authors wish to acknowledge the Shanxi Tianchi Coal Mine for providing the coal specimens used in this study.

**Institutional Review Board Statement:** Not applicable.

**Informed Consent Statement:** Not applicable.

**Data Availability Statement:** Data are available on request due to privacy restrictions.

**Conflicts of Interest:** The authors declare no conflicts of interests.

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
