# Peer review of "Characterization of the Time–Space Evolution of Acoustic Emissions from a Coal-like Material Composite Model and an Analysis of the Effect of the Dip Angle on the Bursting Tendency"

_sustainability, doi:10.3390/su16051711_

Round 1
Reviewer 1 Report
Comments and Suggestions for Authors
Comments to the paper of Pengxiang Zhao, Jian Wen , Shugang Li , Weidong Lu, Yongchen He, Fang Lou and Laolao Wang
“ Characterisation of the time-space evolution on acoustic emission from coal-like material composite model and analysis of the dip angle effect on burst tendency”
The authors studied the influence of the angle of inclination of a coal seam on the strength characteristics, acoustic parameters and damage development when testing model samples consisting of layers of coal and rock under uniaxial pressure conditions. As the inclination angle increased, the damage increased and the assessed characteristics decreased. The authors concluded that the susceptibility to rupture of coal rock decreases as the inclination angle increases. This research result is important both for risk assessment during mine operation and for the development of scientific knowledge on the influence of mixed types of fracture on the strength and acoustic characteristics and development of damage in various materials.
The material presented is thoughtful and well written; the drawings are of good quality. The manuscript is recommended for publication.
Author Response
Dear Reviewer:
Thank you very much for taking the time to review our manuscript. Your feedback has been very helpful for the publication of our article, and we would like to express our gratitude to you once again.
Yours sincerely,
Pengxiang Zhao et al.
Reviewer 2 Report
Comments and Suggestions for Authors
This work investigated the acoustic emission behaviors and fracture modes of the coal-rock materials composite with different coal seam inclination angles. Good experimental results were obtained. The present research is definitely interesting for readers and I believe this work is worth to be published. However, the following comments should be considered before publication.
1. Acoustic emission (AE) technique is an important structural health monitoring method that has been widely used in damage assessment of metal, composite, rock and coal materials. However, it is not clear for readers why the authors used AE technique to investigate the fracture behavior of the coal-rock composite specimens. This should be clarified in the Introduction section.
2. It is necessary to provide the details such as the resonant frequency and operating frequency range of the AE sensor used in this work.
3. Please add a figure or table to summarize the characteristics of different failure modes of specimens with different interface inclination angles (Figure 4).
4. The authors should provide more relevant high-quality English journal papers instead of papers in Chinese (References 2,17,21).
5. The authors extracted energy and count from AE signals to characterize the damage process of specimens with different inclination angles. However, the authors failed to investigate the frequency domain characteristics including the peak frequency, frequency centroid, etc. It has been reported by many investigations (for example, the following references) that the information related to damage mode are generally encoded with the frequency domain of AE waveforms. Therefore, I suggest the authors to extract more frequency features and tie them to different inclination angles and fracture mechanisms. The citation of the following journal papers is also recommended.
https://doi.org/10.1038/s41524-021-00565-xhttps://doi.org/10.1016/j.ijfatigue.2022.106860
6. Does the cumulative AE energy or count show the same decreasing trend with increasing inclination angle In Figure 8?
7. Many grammar errors are found in the manuscript. The English needs an improvement.
Comments on the Quality of English LanguageModerate editing
Author Response
Dear Reviewer:
Thank you very much for taking the time to review our manuscript. Those comments are all valuable and very helpful for revising and improving our paper, as well as the important guiding significance to our researches. We have studied comments carefully and have made correction which we hope meet with approval. Specific modification instructions can be found in the attachment.
Yours sincerely,
Pengxiang Zhao et al.

Reviewer 3 Report
Comments and Suggestions for Authors
This paper study "Analyzing the Temporal and Spatial Changes in Acoustic Emission from a Composite Model of Coal-like Material, and Investigating the Impact of Dip Angle on Burst Tendency", Rock burst poses a serious threat to miners' safety, influenced by coal seam inclination. Unique high-gas coal properties in a Shanxi Province mine prompted safety threshold derivation and testing with various seam inclinations. Tests revealed increased structural damage in coal-rock specimens as inclination rose, with notable effects at 40° and 45° angles. There are still lots of problems that need to be addressed in this study as follows:
1. The introduction part needs to be rewritten, the logic is not good and all the references except one are from the same country, and lots of references are not recently published papers.
2. the authors mention that "The mine coal seam in question is gently inclined (0°–25°)", why did This experiment set seven coal-rock interface inclinations (0°, 10°, 20°, 30°, 35°, 40°, and 45°)?
3. line 121, Weihe sand was aggregated, mass was fixed at 1 200 g, and material percentage (%) was calculated as Weihe sand fixed mass. Does it look like a Google Translate sentence?
4. what does the rock materials component consist of?
5. Section 2.2 is hard to understand, please rewrite it.
6. why the samples' color is different in Figure 4? and if it is developed like the samples in Figure 1? how does the author generate the angle of two materials into one sample and make it reasonable like the field condition?
7. where does the equation 1 to 3 come from?
8. where does the equation 4 come from?
9. the conclusion part is not readable, please rewrite this part.
Comments on the Quality of English Language
Extensive editing of English language required
Author Response

(The authors gave the same response as above.)

Round 2
Reviewer 2 Report
Comments and Suggestions for Authors
After revision, the quality of the manuscript has been greatly improved. I believe it can be published in the journal.
Comments on the Quality of English LanguagePlease further check the grammar errors in the paper.
For example. Line 291 "the frequency characteristics are also an important parameter ".
Author Response

(The authors gave the same response as above.)

Reviewer 3 Report
Comments and Suggestions for Authors
The author responded to some of my questions, but there are still some problems that need to be addressed as follows, and I suggest a minor revision for this paper.
1. The author mentions that "we used the geological conditions of a mine in Shanxi, China as a prototype to make coal rock combination specimens. Although the inclination angle of the coal seam in this mine is between 0 ° and 25 °, due to the complexity of geological conditions and the differences in inclination angles of coal rock interface in each mine, in order to provide certain research ideas for mines with similar geological conditions, this article sets multiple inclination angles of coal rock interface (0 °, 10 °, 20 °, 30 °, 35 °, 40 °, and 45 °) to comprehensively analyze the fracture behavior of coal rock combinations under different inclination angles", it means the author should add more data between 0 ° to 25 °?
2. Considering the deformation and slippage of the slope for the coal-rock model, there can be some modeling work to make the introduction more appealing, such as discrete element method and difference method, some papers may be helpful in the background, identify the Micro-Parameters for Optimized Discrete Element Models of Granular Materials in Two Dimensions Using Hexagonal Close-Packed Structures, FLAC3D-based Analysis on Subgrade Workaround of Asphalt Pavement Structures under Various Loading Conditions
3. The author may also consider preparing a nice picture to show the field condition of the project to make the introduction more appealing. like in this paper, Laboratory Evaluation and Field Demonstration of Cold In-Place Recycling Asphalt Mixture in Michigan Low-Volume Road. a drone picture to show the whole process of the project will be more readable for the reader.
4. The author mentions that "For the rock part of the coal-rock combination specimens, we used sand, cement, and plaster for production. Among them, the mass of sand was 1200g, the mass of cement was 180g, and the mass of plaster was 216g." please add this information to the paper.
Comments on the Quality of English Language
Minor editing of the English language required
Author Response

(The authors gave the same response as above.)
